# Influence of Fatty Acid Desaturase Enzyme-1 Gene (FADS-1) Polymorphism on Serum Polyunsaturated Fatty Acids Levels, Desaturase Enzymes, Lipid Profile, and Glycemic Control Parameters in Newly Diagnosed Diabetic Mellitus Patients

**DOI:** 10.3390/ijms26094015

**Published:** 2025-04-24

**Authors:** Hayder Huwais Jarullah, Eman Saadi Saleh

**Affiliations:** Department of Clinical Laboratory Science, College of Pharmacy, University of Baghdad, Baghdad 10047, Iraq; emansaadi@copharm.uobaghdad.edu.iq

**Keywords:** diabetic mellitus, polyunsaturated fatty acids, desaturase enzymes, genetic polymorphism

## Abstract

Type 2 diabetes mellitus (T2DM) is a prevalent metabolic disorder caused by impaired insulin secretion from pancreatic β-cells and insulin resistance in target tissues. Genome-wide association studies have identified over 50 genetic variants linked to T2DM, including polymorphisms associated with the disease. This study investigates the impact of the *FADS1* (rs174547) polymorphism in T2DM patients compared to healthy controls and examines serum levels of omega-3 and omega-6 fatty acids, as well as D5D and D6D enzyme levels and activity. This case–control study included 120 participants: 60 newly diagnosed T2DM patients and 60 apparently healthy controls matched for age, sex, and other sociodemographic factors. Polyunsaturated fatty acid (PUFA) levels and desaturase enzyme activities in the n-3 and n-6 pathways were assessed using ELISA and gas chromatography. *FADS1* gene polymorphisms were analyzed via Sanger sequencing. Genotype and allele frequencies of *FADS1* (rs174547) differed significantly between groups, with higher frequencies of C-containing alleles in T2DM patients. Multivariate analysis revealed a significant association between the C-allele genotype and increased T2DM risk, independent of sociodemographic variables, lipid profile, and inflammatory markers. In conclusion; reduced serum levels of omega-3 and omega-6 fatty acids in T2DM were associated with decreased desaturase enzyme activity. The *FADS1* (rs174547) polymorphism is significantly associated with T2DM risk, with the minor allele linked to lower desaturase activity.

## 1. Introduction

Type 2 diabetes mellitus (T2DM) is a major public health concern with significant implications for both individual health and healthcare systems. Rapid economic development and urbanization have contributed to a growing global prevalence of diabetes [1,2]. Recent studies report that over one-third of diabetes-related deaths occur in individuals under 60 years of age [3]. In Iraq, the incidence of T2DM has steadily increased, though prevalence rates vary across studies. A 2014 study in Basrah estimated an age-adjusted diabetes prevalence of 19.7%, with 55.7% of cases previously undiagnosed [4]. A 2022 study found that nearly 10% of Iraqis had undiagnosed T2DM and identified associated factors relevant to public health planning [5].

Polyunsaturated fatty acids (PUFAs) are categorized as n-3 (omega-3) or n-6 (omega-6) based on the position of the first double bond relative to the methyl terminus of the fatty acid molecule [6]. The human body cannot synthesize PUFAs with double bonds at carbon positions 3 or 6 from the methyl end; thus, they are considered essential and must be obtained through the diet [7]. Fatty acid synthesis mainly involves the conversion of acetyl-CoA and malonyl-CoA into palmitate, a process requiring nicotinamide adenine dinucleotide phosphate (NADPH) and catalyzed by fatty acid synthase. In mammals, the carbon sources for this process are typically derived from citrate, an intermediate of the tricarboxylic acid cycle. Once synthesized, palmitate can be converted into various long-chain fatty acids used in membrane biosynthesis and lipid droplet formation [8]. Fatty acid synthesis includes the de novo generation of saturated fatty acids from acetate and the elongation and desaturation of key dietary fatty acids—linoleic acid (LA; C18:2n-6) and alpha-linolenic acid (ALA; C18:3n-3)—into highly unsaturated 20- and 22-carbon fatty acids [9].

Previous studies have demonstrated that plasma phospholipid levels of long-chain polyunsaturated fatty acids (LC-PUFAs) are involved in inflammation [10], obesity [11], and type 2 diabetes [12]. LC-PUFAs can be obtained from the diet or synthesized endogenously from essential fatty acids—linoleic acid (LA) and alpha-linolenic acid (ALA) [13]. Humans cannot produce these essential fatty acids due to the absence of enzymes needed to introduce double bonds beyond the delta-9 carbon position; therefore, they must be consumed through dietary sources [13]. Linoleic acid (18:2, n-6) and ALA (18:3, n-3) belong to the omega-6 and omega-3 fatty acid families, respectively [14]. Following ingestion, both fatty acids are metabolized via a common enzymatic pathway. Two enzyme families—elongases and desaturases—play pivotal roles in converting LA and ALA into bioactive LC-PUFAs [15].

Polyunsaturated fatty acids (PUFAs), including arachidonic acid (AA; C20:4n-6), eicosapentaenoic acid (EPA; C20:5n-3), and docosahexaenoic acid (DHA; C22:6n-3), are precursors to lipid mediators such as eicosanoids and pro-resolving molecules. Eicosanoids, which are 20-carbon derivatives of AA, dihomo-γ-linolenic acid (C20:3n-6), and EPA, are produced through cyclooxygenase and lipoxygenase pathways, resulting in prostaglandins and leukotrienes, respectively. While eicosanoids are primarily pro-inflammatory, some lipid mediators, such as 15-deoxy-Δ12,14-PGJ2 (15d-PGJ2), exhibit anti-inflammatory properties. EPA and DHA are also metabolized into pro-resolving mediators—such as lipoxins and resolvins—via lipoxygenases, cytochrome P450, and acetylated cyclooxygenase-2. Notably, free PUFAs, particularly AA, can activate peroxisome proliferator-activated receptors (PPAR)-α and -γ [16].

Desaturases are key enzymes in lipid metabolism, significantly influencing the desaturation index, which is closely associated with an elevated risk of insulin resistance (IR) [17]. These enzymes introduce double bonds into fatty acid carbon chains, altering their structure [18]. This desaturation impacts the physical properties of fatty acids, including membrane fluidity and intracellular signaling pathways [19]. Δ6-desaturase (D6D) and Δ5-desaturase (D5D) are critical for long-chain polyunsaturated fatty acid (LC-PUFA) synthesis. D6D acts on linoleic acid (LA) and alpha-linolenic acid (ALA), inserting a double bond at the Δ6 position to form γ-linolenic acid and stearidonic acid (SDA), respectively. D5D subsequently introduces a double bond at the Δ5 position, producing arachidonic acid and eicosapentaenoic acid (EPA), which are precursors to bioactive lipid mediators [15]. Desaturase activity strongly influences membrane composition and lipid-mediated signaling, thereby affecting various physiological functions. Altered desaturase function has been linked to metabolic disorders, cardiovascular diseases, and inflammatory conditions [15].

A locus on human chromosome 11, identified as q12–q13, contains the FADS1 (fatty acid desaturase-1), FADS2 (fatty acid desaturase-2), and FADS3 (fatty acid desaturase-3) genes. These genes share similar exon–intron structures and are clustered together, likely due to gene duplication events [20]. Genetic variations, particularly single-nucleotide polymorphisms (SNPs) in FADS1 and FADS2, can influence D5D and D6D enzyme activity, leading to changes in blood fatty acid composition and lipid profiles. Some minor alleles in these genes are linked to reduced D5D activity and increased D6D activity, thereby altering LC-PUFA levels—such as arachidonic acid—and affecting lipid metabolism and disease risk [21]. While earlier studies showed that FADS3 could synthesize sphinganine ceramides using sphingosine ceramides as substrates, other findings suggest it also acts on free sphingosine. Thus, the specific substrates of FADS3 remain debated [22].

Although several studies have investigated FADS1 polymorphisms in general populations and various diseases, limited research has explored its association with diabetes. Moreover, no studies have examined the effect of FADS1 variants in Middle Eastern populations, representing a key knowledge gap addressed in this study. This research focuses on FADS1 gene polymorphisms and associated differences in plasma enzyme levels. It is the first study conducted in the Iraqi population to compare individuals newly diagnosed with type 2 diabetes with healthy controls. The findings aim to clarify the distribution of FADS1 polymorphisms in these groups and their impact on plasma glucose and lipid levels.

## 2. Results

### 2.1. Assessment of Various Variables and Biochemical Markers in the Study Groups

This study included 120 participants, comprising 60 healthy controls and 60 patients with T2DM. No significant differences were observed between groups in age (45.6 ± 6.7 vs. 47.8 ± 9.9 years), sex distribution (male-to-female ratio: 1.22:1 vs. 1.87:1), BMI (27.2 ± 2.1 vs. 27.0 ± 2.0 kg/m^2^), WHR (0.837 ± 0.041 vs. 0.841 ± 0.039), education levels, or dietary habits. Median fasting plasma glucose (FPG) (98 vs. 136.5 mg/dL), fasting insulin (8.6 vs. 10.21 µU/mL), HOMA-IR (2.2 vs. 3.53), and mean HbA1c (5.73 vs. 9.50%) were significantly lower in controls than in T2DM patients. No significant differences were found in plasma levels of Apo A, Apo B, Apo B/A ratio, total cholesterol, or LDL-C. However, triglycerides, VLDL, and hs-CRP levels were significantly higher, and HDL-C significantly lower, in T2DM patients. Serum levels of all omega-3 PUFAs (ALA, SDA, ETA, EPA, DHA) and omega-6 PUFAs (LA, GLA, DGLA, AA) were also significantly lower in T2DM patients compared to controls, as depicted in Table 1.

The plasma levels of D5D and D6D were slightly lower in T2DM patients than in the controls, though the differences were not statistically significant. The estimated D5D activity in the omega-3 pathway was significantly higher in T2DM patients and marginally elevated in the omega-6 pathway. Conversely, the estimated D6D activity was significantly reduced in the omega-3 pathway and slightly lower in the omega-6 pathway among T2DM patients. Elongase-5 activity was significantly increased in the omega-3 pathway and significantly decreased in the omega-6 pathway in T2DM patients, as depicted in Figure 1.

### 2.2. Genetic Analysis

The studied SNP (rs174547) conformed to Hardy–Weinberg equilibrium. In FADS1 (rs174547), both genotype and allele frequencies differed significantly between the controls and T2DM patients, with higher frequencies of C-containing alleles and genotypes observed in the T2DM group. As shown in Table 2, genotypes containing the C allele (CC or TC) were associated with an increased risk of T2DM. In the co-dominant model, the TC genotype was linked to a 3.176-fold higher risk compared to the TT genotype. Additionally, the combined TT + TC genotypes showed a 3.051-fold increased risk of T2DM relative to TT alone.

### 2.3. Multivariate Analysis

The presence of the C-allele genotype in FADS1 (rs174547) was associated with an increased risk of T2DM. This association remained significant after adjusting for sociodemographic factors, lipid profile parameters, and inflammatory markers, indicating that the rs174547 polymorphism is independently linked to T2DM, as depicted in Table 3.

### 2.4. Assessment of Lipid Profile, Apolipoproteins, and Inflammatory Mediator in Type 2 Diabetes Patients Under the Effect of rs174547 Polymorphism

In T2DM patients, the rs174547 polymorphism was not associated with significant differences in FPG, HbA1c, fasting insulin, or HOMA-IR values. However, serum cholesterol and LDL-C levels were influenced by genotype, with the TC genotype showing significantly lower levels than both CC and TT genotypes. No significant differences were observed between the CC and TT genotypes for these parameters. Other variables, including Apo A, Apo B, Apo B/A ratio, HDL-C, triglycerides, VLDL, and hs-CRP, were not significantly affected by rs174547 polymorphism, as depicted in Figure 2.

The rs174547 polymorphism influenced serum cholesterol and LDL-C levels, with the TC genotype showing significantly lower values than both CC and TT genotypes. No significant differences were observed between CC and TT genotypes. Other parameters, including Apo A, Apo B, Apo B/A ratio, HDL-C, triglycerides, VLDL, and hs-CRP, were not significantly affected by rs174547 polymorphism, as illustrated in Figure 3.

### 2.5. Assessment of Plasma Polyunsaturated Fatty Acids in Type 2 Diabetes Patients Under the Effect of rs174547 Polymorphism

Plasma ALA levels were significantly higher in T2DM patients with the TC genotype compared to those with the TT genotype. However, rs174547 polymorphism was not associated with significant differences in the remaining omega-3 and omega-6 PUFAs (SDA, ETA, EPA, DHA, LA, GLA, DGLA, and AA) in T2DM patients, as depicted in Figure 4.

### 2.6. Assessment of Desaturase Enzymes in Type 2 Diabetes Patients Under the Effect of rs174547 Polymorphism

In T2DM patients, rs174547 polymorphism was not associated with significant differences in D5D or D6D enzyme levels. Similarly, no significant differences were observed in the SDA/ALA, ETA/SDA, EPA/ETA, GLA/LA, DGLA/GLA, or AA/DGLA ratios, as presented in Table 4.

## 3. Discussion

In this study, the mean age of newly diagnosed T2DM patients was 47.8 ± 9.9 years, consistent with both international and local research. A large study across 19 high-income countries reported a mean diagnostic age of 54.4 ± 9.8 years [23], while a U.S. study of 8654 patients reported 51.71 years [24], and a Chinese study reported 52.91 ± 10.25 years [25]. In Iraq, a Basrah study involving over 5400 patients found a mean age of 46.7 ± 14.3 years [4]. Two Baghdad studies reported means of 49.17 ± 7.31 and 45.3 ± 5.67 years, respectively [26,27]. These international and local findings align with the present study [28,29,30,31].

The ω-3 and ω-6 PUFA are recognized as important immune nutrients with pharmacological effects, including lipid regulation, inflammation reduction, and immune enhancement. However, their role in type 2 diabetes remains debated [32,33]. Some studies suggest that ω-3 PUFA supplementation offers protective effects against various complications associated with type 2 diabetes [34,35,36].

The rs174547 variant is a key regulator of ω-6 PUFA, closely associated with its synthesis and serum levels [37]. In this study, the C allele of FADS1 (rs174547) was significantly associated with T2DM and was more prevalent in T2DM patients (T/C: 0.792/0.208) than in controls (T/C: 0.908/0.092). Genotype distribution also differed notably between groups, with TT being most common (TT/TC/CC: 0.65/0.283/0.067 in T2DM vs. 0.85/0.117/0.033 in controls).

Globally, the T/C allele frequency of the rs174547 SNP is approximately 0.673/0.327. However, distribution varies across ethnic groups, with African populations showing the highest T/C ratio (0.921/0.079), followed by South Asians (0.821/0.179), Europeans (0.676/0.324), and East Asians (0.608/0.392) [38]. In this study, the T/C ratio in the control group resembled that of African populations, whereas the T2DM group was more similar to South Asian populations.

Only one study has examined the rs174547 genotype specifically in T2DM. Tiwana’s study of 820 T2DM patients reported genotype frequencies of TT/TC/CC as 0.341/0.481/0.174, with a T/C allele frequency of 0.583/0.417—showing a higher rate of heterozygosity and a lower frequency of both the wild-type TT genotype and T allele compared to the current study [39]. Other research has reported rs174547 distributions in different populations and conditions. In healthy Japanese lactating women, the genotype distribution was 0.375/0.497/0.128 (TT/TC/CC) [40,41]. A post hoc GWAS analysis from the PREDIMED Plus-Valencia study showed a T/C allele frequency of 0.702/0.298 [42]. In metabolic syndrome, the genotype distribution was 0.468/0.426/0.106, with a T/C ratio of 0.681/0.319 [43]. Among healthy Malaysian participants, the distribution was 0.395/0.320/0.285, with T/C frequencies of 0.555/0.445 [44]. In contrast, the current study observed a higher TT-to-TC ratio than previously reported, particularly in Iraqi T2DM patients, where TC genotype frequency was lower in T2DM than in controls (28.3% vs. 11.7%).

An exploratory multivariate analysis was conducted to examine the relationship between the rs174547 genotype (dominant model) and T2DM risk. After adjusting for sociodemographic factors, serum lipid profile, and hs-CRP, the CC + TC genotype was independently associated with an increased risk of T2DM. However, this association lost statistical significance after further adjustment for serum desaturase (δ5 and δ6) enzyme levels, suggesting that the link between rs174547 and T2DM is influenced by desaturase levels. As existing literature primarily focuses on desaturase activity, measured via product-to-substrate ratios rather than direct enzyme levels as in this study, comparison with previous findings remains challenging. Notably, identifying an independent association between the CC + TC genotype and T2DM represents a novel finding.

Some studies have reported opposing results, suggesting that ω-3 PUFAs have either no effect or a potentially harmful impact on diabetes risk [45,46]. The ASCEND trial, involving approximately 15,000 T2DM patients, found that daily supplementation with 1 g of ω-3 PUFA did not significantly affect cardiovascular events or all-cause mortality compared to 1 g of olive oil over a 7.4-year follow-up [47]. Additional evidence from clinical trials and cohort studies suggests that ω-3 PUFA intake—whether from supplements or food—has neither beneficial nor adverse effects on type 2 diabetes [48,49]. While these studies address dietary PUFAs and T2DM risk, the present study directly investigated the impact of PUFAs in relation to FADS1 polymorphism. A similar association was reported in a study on a different ethnic group (Tiwana’s study) [39].

The association between ω-3 PUFA consumption and T2DM is influenced by both genetic and non-genetic factors. Polymorphisms in the FADS gene family affect the biological role of ω-3 PUFAs in T2DM. D5D and D6D, enzymes involved in ω-3 PUFA metabolism, are encoded by FADS1 and FADS2, respectively. Genetic variation within these genes is linked to the reduced expression and activity of these enzymes, thereby altering ω-3 PUFA concentrations [50].

A pharmacokinetic study of 161 T2DM patients established a dose–response relationship between plasma ω-3 PUFA levels and HbA1c using nonlinear mixed-effect analysis, highlighting the complexity of this interaction [32]. The study found that HDL significantly influenced this relationship: individuals with lower HDL exhibited faster ω-3 PUFA clearance, reducing the effectiveness in lowering HbA1c. This underscores the critical role of HDL levels in HbA1c regulation [32]. A cross-sectional study in T2DM patients aged ≥60 years identified a U-shaped relationship between HDL-C and HbA1c, with an inflection point at 60 mg/dL [51]. Dietary PUFAs, especially ω-3 PUFAs, promote the transport of HDL-bound apoE, which supports reverse cholesterol transport through mechanisms such as HDL subspecies secretion, cholesterol efflux from macrophages, HDL enlargement via cholesterol ester uptake, and rapid clearance. These effects help reduce T2DM risk. Conversely, apoC3 counteracts the benefits of apoE, possibly explaining the link between HDL-apoC3 and T2DM [52]. In this study, HDL levels were significantly lower in T2DM patients, aligning with Wang et al. [32], further supporting the complex interaction between PUFAs, HDL, and T2DM risk. However, rs174547 polymorphism did not affect HDL-C levels, suggesting that the HDL-T2DM relationship is independent of FADS1 genotype.

PUFAs are vital components of cellular membranes, incorporated into phospholipids and other complex lipids through esterification to glycerol or other polyols. These lipids often contain hydrophilic groups that enhance the structural and functional integrity of cell membranes [53,54,55]. Saturated fats, with their straight-chain structure, create tightly packed and rigid lipid bilayers. In contrast, PUFAs, due to their multiple double bonds and bent structure, promote a more fluid and dynamic membrane environment. This variation in lipid composition directly affects membrane fluidity, which is essential for normal cellular functions such as receptor mobility, intracellular signaling, and nutrient and ion transport. The accumulation of saturated fats can stiffen membranes, reducing insulin receptor mobility and impairing interactions with signaling proteins. This disruption in insulin signaling contributes to insulin resistance, a key factor in the development of T2DM [56,57].

Omega-3 fatty acids play a key role in maintaining insulin sensitivity and regulating glucose metabolism. They enhance insulin action in target tissues, such as muscle and adipose tissue, by strengthening signaling pathways associated with insulin receptors. Omega-3s also promote the expression of glucose transporter type 4 (GLUT4) through mechanisms involving peroxisome proliferator-activated receptor gamma (PPAR-γ), a critical transcription factor that improves insulin responsiveness. A diet low in omega-3 and high in omega-6 fatty acids disrupts the fatty acid balance in cell membranes, leading to increased production of pro-inflammatory mediators from omega-6 fatty acids. This imbalance exacerbates systemic inflammation and insulin resistance. Chronic inflammation impairs insulin signaling, reduces glucose uptake, and increases the risk of developing type 2 diabetes [58].

Omega-3 fatty acids regulate triglyceride levels by modulating the activity of several nuclear receptors, including sterol regulatory element-binding protein (SREBP), liver X receptor-alpha (LXRα), retinoid X receptor-alpha (RXRα), farnesoid X receptor (FXR), and peroxisome proliferator-activated receptors (PPARs) [59]. SREBPs are transcription factors that control genes involved in lipid synthesis and are implicated in both physiological and pathological processes [60]. Omega-3s reduce hepatic lipid production by inhibiting SREBP-1c activity, thereby decreasing VLDL formation, fatty acid synthesis, and triglyceride-producing enzymes [61]. They suppress SREBP-1c expression by preventing the LXR/RXR heterodimer from binding to LXR response elements (LXREs) within the SREBP-1c promoter, a critical step for gene activation [62]. Lipid accumulation in non-adipose tissues is strongly linked to T2DM and its complications. Increased SREBP-1c expression has been observed in the liver and islets of diabetic rats [63], suggesting that lower omega-3 levels may elevate SREBP activity and T2DM risk.

Endoplasmic reticulum (ER) stress also plays a key role in insulin resistance. Imbalances in fatty acid composition can induce ER stress, triggering the unfolded protein response (UPR) and impairing insulin signaling. The interplay between fatty acid composition, inflammation, and cellular stress underscores the importance of maintaining a balanced PUFA intake in preventing and managing insulin resistance and T2DM. Omega-3 fatty acids enhance insulin sensitivity through PPAR-γ activation and anti-inflammatory effects, whereas excessive omega-6 intake promotes insulin resistance via inflammation, dysregulated adipokine release, and ER stress induction [64,65].

Genetic studies on PUFAs have identified associations with SNPs in FADS1 and FADS2. Human desaturase research typically estimates enzyme activity using product-to-precursor fatty acid ratios. Since most circulating lipids during fasting are liver-derived, desaturation indices from fasting samples are considered reflective of hepatic desaturase activity [66,67]. In addition to enzymatic activity, the present study directly measured serum desaturase enzyme levels.

In this study, D5D enzyme levels were higher in participants with the TC genotype of rs174547 compared to other genotypes, although D5D activity was lower in this group. However, these differences were not statistically significant. D6D enzyme levels and activity also showed no significant variation by rs174547 genotype, as illustrated in Figure 5.

In Huang et al. (2017), among T2DM patients, the TC genotype of rs174547 was associated with lower D5D activity compared to TT and similar activity to CC. The study also reported significant differences in D6D activity, with the highest levels in TT, followed by TC and lowest in CC genotypes [39]. These findings partially align with the present study regarding D5D activity, but differ concerning D6D activity. Potential explanations include the limited number of participants with CC and TC genotypes in the current study, ethnic differences between populations, and variation in disease stage. The present study examined newly diagnosed T2DM patients in their 5th decade of life, whereas Huang et al. (2017) focused on patients with established T2DM in their 6th decade [39]. These factors may account for the discrepancies.

In a study on healthy Japanese individuals, the rs174547 SNP had no effect on D5D activity but significantly influenced D6D activity, with the highest levels observed in TT, followed by TC and CC genotypes [68]. Similarly, a Polish study involving postmenopausal women found no significant effect on D6D activity, while D5D activity was significantly higher in the TT genotype compared to TC + CC genotypes [69]. Previous research suggests that the FADS1 genotype modulates systemic FADS1 activity, with the minor C allele associated with reduced D5D activity [37,70]. A study on calcified and non-calcified aortic valve tissue also reported that the TT genotype exhibited the highest D5D and D6D activity, followed by TC and CC [71]. One possible explanation is that the C allele of rs174547 impairs desaturase function, leading to the reduced synthesis of long-chain n-6 and n-3 PUFAs, particularly 20:4n-6 and 20:5n-3 [72].

In this study, D5D and D6D enzyme levels were lower in T2DM patients, whereas D5D and D6D enzymatic activities were higher. A Mendelian randomization study suggested that genetically reduced D5D activity is associated with increased diabetes risk [73]. However, the molecular mechanisms linking D5D and D6D to diabetes remain poorly understood. Cross-sectional studies have linked altered D5D and D6D activity to insulin resistance [74,75], while lifestyle-driven changes in desaturase activity have been associated with shifts in insulin sensitivity [76]. The rs174550 genotype in FADS1—fully linked with rs174546—was associated with early insulin secretion but not insulin sensitivity [77], and another study found no consistent relationship between this genotype and β-cell function [78]. It is likely that altered fatty acid composition mediates the impact of desaturase activity on diabetes risk, as membrane lipid content may influence insulin signaling, receptor binding, and function. Additionally, LC-PUFAs can act as ligands for transcription factors such as sterol regulatory element-binding protein 1 and peroxisome proliferator-activated receptors [79,80,81].

In recent decades, many Middle Eastern and North African countries have experienced a dietary shift from traditional plant- and seafood-rich diets to more Westernized diets high in saturated and trans fats. This transition, driven by urbanization, economic growth, and increased access to processed foods, has contributed to rising rates of obesity, diabetes, and cardiovascular disease—often compounded by sedentary lifestyles and limited preventive care access [82]. This shift may explain the lower PUFA levels observed in T2DM patients.

The present findings offer insights into omega-3 supplementation strategies. Diabetic patients exhibited reduced serum PUFA levels, alongside decreased D5D and D6D levels and activity. Moreover, individuals with the TT genotype of rs174547 had lower enzyme levels and activity, suggesting a need for higher supplementation with LA and ALA.

## 4. Methods

### 4.1. Study Design and Settings

This case–control study included 120 participants divided into two groups: 60 individuals newly diagnosed with T2DM and 60 apparently healthy controls, matched for age, sex, and other relevant sociodemographic variables. T2DM diagnosis followed the 2024 American Diabetes Association guidelines, based on either FPG > 126 mg/dL and/or HbA1c > 6.5% [83]. Participants were recruited from the Diabetes and Endocrinology Center, Nasiriyah, Dhi-Qar Health Directorate, Iraqi Ministry of Health. The study was conducted from 5 March to 25 October 2024, and adhered to the STROBE guidelines for reporting observational studies [84,85].

### 4.2. Ethical Approval

Ethical approval was obtained from the Scientific and Ethical Committee, College of Pharmacy, University of Baghdad (Approval No. REC042024191H), on 1 March 2024. All participants were informed about the study’s purpose and potential benefits, and written informed consent was obtained prior to enrollment.

### 4.3. Patient Selection and Data Collection

Following consultation with the center’s endocrinologist regarding inclusion and exclusion criteria, eligible patients were referred to the primary investigator. Relevant data, including age, sex, diet, education level, BMI, and waist-to-hip ratio, were recorded. A venous blood sample (10 mL) was collected from each participant for analysis. All examinations were provided free of charge, with no financial burden on the participants.

#### 4.3.1. Inclusion Criteria

Eligible participants were aged 18 years or older, of either sex, not taking any medications or supplements, and free of severe chronic diseases other than diabetes in the T2DM group.

#### 4.3.2. Exclusion Criteria

Exclusion criteria included obesity (BMI ≥ 30 kg/m^2^), waist-to-hip ratio > 0.85 in females or > 0.9 in males, pregnancy, current or former smoking, and alcohol consumption.

### 4.4. Obesity Assessment

Each participant’s weight, height, waist circumference, and hip circumference were measured. BMI was calculated by dividing weight in kilograms by the square of height in meters [86], with obesity defined as BMI ≥ 30 kg/m^2^ [87]. Waist-to-hip ratio (WHR) was determined by dividing waist circumference by hip circumference [88]. According to WHO criteria, WHR > 0.85 in females and >0.9 in males was classified as abdominal obesity [89].

### 4.5. Specimen Collection

Approximately 10 mL of venous blood was collected from each participant. Of this, 1.5 mL was placed in an EDTA tube; 0.5 mL was used for HbA1c measurement, while the remaining 1.0 mL was frozen for subsequent DNA extraction [90]. The remaining 8 mL of blood was transferred to a gel tube and left at room temperature for 30 min to allow clotting, followed by centrifugation at 4000 rpm for 10 min to separate the serum. The extracted serum was divided into two portions: one was used immediately for measuring glucose, insulin, hs-CRP, Apo A, Apo B, and the lipid profile; the second was stored in Eppendorf tubes at −40 °C for later ELISA and gas chromatography (GC) analysis, as depicted in Figure 6.

### 4.6. Routine Laboratory Biochemical Analysis

#### 4.6.1. Plasma Glucose Assessment

Serum glucose levels were measured using the glucose hexokinase GLUC3 kit (Roche, Mannheim, Germany), designed for the Cobas® c 311 analyzer (Roche, Mannheim, Germany). This enzymatic assay involves the hexokinase-mediated phosphorylation of glucose to glucose-6-phosphate, which is subsequently converted to 6-phosphogluconate, producing nicotinamide adenine dinucleotide (NADH). The amount of NADH formed is directly proportional to the glucose concentration (mg/dL) and is measured photometrically at 459 nm [91].

#### 4.6.2. Glycated Hemoglobin Assessment

HbA1c levels were measured using the A1C3 kit (Roche, Mannheim, Germany) on the Cobas® c 311 analyzer (Roche, Mannheim, Germany). Tetradecyltrimethylammonium bromide (TTAB) was used in the hemolyzing reagent to eliminate leukocyte interference. HbA1c was determined as a percentage using a turbidimetric inhibition immunoassay (TINIA) on hemolyzed whole blood. Glycohemoglobin in the sample reacted with anti-HbA1c antibodies to form soluble antigen–antibody complexes [92].

#### 4.6.3. Serum Insulin Assessment

Serum insulin levels (µU/mL) were quantified using the Insulin CalSet kit (Roche, Mannheim, Germany) on the Cobas e411 analyzer (Roche, Mannheim, Germany), employing electrochemiluminescence immunoassay (ECLIA) technology [93]. A 20 µL sample was incubated with two monoclonal insulin-specific antibodies—one biotinylated and the other labeled with a ruthenium complex—forming a sandwich complex. Streptavidin-coated microparticles were added to anchor the complex via biotin–streptavidin interaction. The reaction mixture was transferred to a measuring cell, where microparticles were magnetically captured onto the electrode surface. After the removal of unbound substances, a voltage induced chemiluminescence, which was measured by a photomultiplier at 450 nm. The emitted light intensity was directly proportional to the insulin concentration [94].

#### 4.6.4. Homeostatic Model Assessment for Insulin Resistance

The Homeostasis Model Assessment (HOMA) is a mathematical framework for evaluating IR based on fasting glucose and insulin levels [95]:HOMA−IR=[Fasting Insulin(μU/mL) × FPG(mg/dL)]÷405

#### 4.6.5. High-Sensitivity C-Reactive Protein Assessment

The CRP4 kit, designed for use with Cobas^®^ c 311 analyzers, was used to measure serum CRP levels (mg/L). Prepared serum samples were loaded into the analyzer, which automatically performed the test using a particle-enhanced turbidimetric method. In this assay, human CRP agglutinates with latex particles coated with anti-CRP antibodies, leading to increased turbidity. The resulting turbidity was measured photometrically at 450 nm [96].

#### 4.6.6. Lipid Profile Assessment

The current study employed the TRIGL kit (Roche, Mannheim, Germany) on the Cobas^®^ c 311 analyzer to measure total triglycerides (mg/dL) using an enzymatic colorimetric assay [97]. Total cholesterol was assessed with the CHOL2 kit [98], HDL with the HDLC4 kit [99], and LDL with the LDLC3 kit [100], all using the same analyzer and assay principle. These enzymatic colorimetric assays were based on absorbance readings at 450 nm [101,102,103,104,105].

Very-low-density lipoprotein (VLDL) (mg/dL) was calculated by dividing triglycerides by five [106].VLDL−C=[Triglycerides]/5

#### 4.6.7. Lipoprotein Assessment

Apolipoprotein A and B (Apo A and Apo B) levels (g/L) were measured using an immunoturbidimetric assay on the Cobas® c 311 analyzer. In this method, anti-apolipoprotein antibodies react with serum antigens, forming antigen–antibody complexes upon agglutination. These complexes are quantified using turbidimetric analysis [107,108].

#### 4.6.8. Enzyme-Linked Immunosorbent Assay of Omega-6 Fatty Acids and Desaturase Enzyme Levels

Enzyme-linked immunosorbent assay (ELISA) kits (Yl Biont, Shanghai, China) were used to quantify serum levels of linoleic acid (LA) and its physiologically active omega-6 derivatives, including Gamma-linolenic acid (GLA), dihomo-Gamma-linolenic acid (DGLA), and arachidonic acid (AA). Serum levels of delta-5-desaturase (D5D) and delta-6-desaturase (D6D) enzymes were also measured using this method [109,110]. Quantification was performed using a microplate reader (ELISA system, Biotek 800 TS, Winooski, VT, USA).

All kits employed biotin double-antibody sandwich ELISA technology. Fatty acids from samples and standards were added to wells pre-coated with fatty acid monoclonal antibodies. After incubation, biotin-labeled anti-fatty acid antibodies and streptavidin–HRP were added, forming an immune complex. Following incubation and washing, substrates A and B were introduced, producing a color change from blue to yellow upon the addition of the stop solution. The intensity of the final color was positively correlated with fatty acid concentration [111].

#### 4.6.9. Serum Omega-3 Fatty Acid Measurements

Gas chromatography (GC) was used to determine serum levels of the essential omega-3 fatty acid alpha-linolenic acid (ALA) and its physiologically active derivatives, including stearidonic acid (SDA), eicosatetraenoic acid (ETA), eicosapentaenoic acid (EPA), and docosahexaenoic acid (DHA). A 200 µL serum sample was used for fatty acid extraction. To each sample, 50 µL of 0.05% sulfuric acid (H_2_SO_4_) was added to lower the pH and enhance extraction efficiency from serum proteins. The mixture was vortexed for at least 30 s, followed by the addition of 2 mL ethyl acetate, a polar solvent used to extract fatty acids from the aqueous phase. After vortexing for 60 s, the sample was centrifuged at 4000 rpm for 10 min at 4 °C to separate the ethyl acetate (organic) phase. The organic layer, containing the fatty acids, was collected and evaporated to dryness under a nitrogen stream.

The residue was reconstituted in 2 mL of an H_2_SO_4_–methanol–toluene mixture (5:90:5, *v*/*v*) to initiate transesterification. In this reaction, H_2_SO_4_ acted as a catalyst, methanol (CH_3_OH) as the methylating agent to form fatty acid methyl esters (FAMEs), and toluene as a stabilizer to improve FAME solubility. The mixture was incubated at 75 °C for 1 h with intermittent shaking every 20 min to complete the transesterification process [112].

Following incubation, the sample was cooled to room temperature, and 1 mL of saturated NaCl solution followed by 2 mL of hexane was added. NaCl enhances FAME partitioning into the hexane phase by reducing the solubility of polar compounds, while hexane, a non-polar solvent, efficiently extracts FAMEs from the aqueous layer. The mixture was vortexed for 60 s to ensure thorough mixing and FAME transfer into the organic phase. After centrifugation, the upper hexane layer, containing the FAMEs, was collected and evaporated to dryness under a nitrogen stream. The residue was re-dissolved in 200 µL of n-hexane for gas chromatography analysis and filtered through a 0.25 µm membrane to remove particulates that could interfere with chromatography [112].

A non-polar to slightly polar capillary column (DB-5; 30 m length, 0.25 mm internal diameter) was used to separate fatty acid methyl esters (FAMEs) based on chain length and degree of unsaturation. Nitrogen served as the carrier gas at a flow rate of 100 kPa. FAMEs were analyzed using a Shimadzu GC-2010 gas chromatography system (Shimadzu Corporation, Kyoto, Japan) equipped with a flame ionization detector (FID). Retention times were compared with known standards to identify individual fatty acids.

A calibration curve was prepared using known standards of ALA, SDA, ETA, eicosapentaenoic acid (EPA), and docosahexaenoic acid (DHA). The peak area of each fatty acid methyl ester in the sample chromatogram was integrated, and concentrations were calculated by comparison with the calibration curve. Chromatographic data were analyzed using IC Solution software (version 2.4), and the results were expressed as the weight percent of total fatty acids. Fatty acid concentrations were reported as a percentage of the total peak area (weight%) [39].

#### 4.6.10. Estimation of Desaturase and Elongase Enzyme Activity for Omega-3 and -6 Pathways

Based on the product-to-substrate ratio, the enzyme activity of D5D, D6D, and elongase enzymes was estimated as follows for the omega-3 pathway [113,114]:D5D=EPA/ETAElongase=ETA/SDAD6D=SDA/ALA

While for the omega-6 pathway [113,114]:D5D=AA/DGLAElongase=DGLA/GLAD6D=GLA/LA

### 4.7. Genomic Examination

For DNA extraction, frozen blood samples collected in EDTA tubes and stored at –40 °C were used. Genomic DNA was isolated using the EasyPure^®^ Blood Genomic DNA Kit (TransGen Biotech, EE121-02, Beijing, China), following the manufacturer’s instructions. The protocol involves cell lysis, protein removal, DNA binding, washing, and elution, providing a simple and efficient method for extracting high-quality genomic DNA from 5 to 250 µL of frozen blood [115,116,117].

To extract genomic DNA, 20 µL of proteinase K, 500 µL of BB3 buffer, and 20 µL of RNase were added to a microcentrifuge tube and vortexed for 15 s. The mixture was incubated at room temperature for 10 min, followed by centrifugation. The lysate was transferred to a spin column and centrifuged at 12,000 rpm for 1 min; the flow-through was discarded. Next, 500 µL of ethanol-containing wash buffer (95–100%) was added, and the column was centrifuged at 12,000 rpm for 30 s. This washing step was repeated twice to ensure the removal of residual contaminants. The spin column was placed into a collection tube, centrifuged at 12,000 rpm for 2 min, and then air-dried at room temperature. To elute DNA, the column was transferred to a 1.5 mL microcentrifuge tube, and 50–200 µL of elution buffer (prewarmed to 60 °C) or distilled water (pH > 7.0) was added to the column center. After 1 min of incubation at room temperature, the column was centrifuged at 12,000 rpm for 1 min. DNA presence and integrity were evaluated using 1% agarose gel electrophoresis. Ten microliters of each DNA sample were run on the gel stained with ethidium bromide and visualized under a UV transilluminator after applying a 100-volt current for 1 h [118,119].

The purity and concentration of the extracted genomic DNA were assessed using a NanoDrop spectrophotometer (Thermo Fisher Scientific, Wilmington, DE, USA). Absorbance was measured at 260 nm and 280 nm, with DNA absorbing at 260 nm and protein contaminants at 280 nm. DNA purity was determined by the A260/A280 ratio, with values between 1.7 and 2.0 indicating good quality suitable for PCR applications [120,121,122]. Polymerase chain reaction (PCR) was used to amplify the isolated genomic DNA.

Primers were designed using Primer3Plus [123,124] and verified using UCSC tools [125] and reference sequences from the National Center for Biotechnology Information (NCBI), as shown in Appendix A. The FADS1 (rs174547) primers were forward (F) 5′-AGGAGTTGGCTTGGGAAAGT-3′ and reverse (R) 5′-TCTCTGCTCCCACCTGTACC-3′, with a primer size of 20 bp and a product size of 604 bp. For each assay, primers were prepared by dissolving lyophilized samples in nuclease-free water per the manufacturer’s instructions (Alpha DNA Ltd., Montreal, QC, Canada). A 100 μM stock solution was prepared and stored at −20 °C. A 10 μM working solution was obtained by diluting 10 μL of stock with 90 μL of nuclease-free water and stored at −20 °C until use.

A partial sequence was selected to investigate the association between FADS1 gene polymorphism and T2DM in Iraqi patients [116]. PCR optimization involved testing four annealing temperatures: 56, 58, 60, and 62 °C. An annealing temperature of 58 °C produced the clearest and most distinct bands on agarose gel and was therefore used in this study [126]. PCR was performed using 2xEasyTaq^®^ PCR SuperMix (TransGen Biotech, Beijing, China) in a final volume of 25 μL, following the manufacturer’s instructions.

Extracted DNA and PCR-amplified fragments were separated by agarose gel electrophoresis and visualized under UV light following ethidium bromide staining [126]. A 1% (*w*/*v*) agarose gel was used for DNA extraction, and a 2% (*w*/*v*) gel for PCR product detection. One gram of agarose was dissolved in 100 mL of 1X TBE buffer (0.89 M Tris, 0.89 M boric acid, 2 mM EDTA; pH = 8) with continuous heating and stirring. After cooling to 60 °C, ethidium bromide was added to a final concentration of 0.5 µg/mL. The agarose was poured into a casting tray, and a comb was inserted. Once solidified, the comb was carefully removed. The gel tray was placed in a horizontal electrophoresis tank and covered with 1X TBE buffer (3–5 mm above the gel surface). Each well was loaded with 10 µL of DNA or PCR product, along with 5 µL of a 100 bp DNA ladder. Electrophoresis was conducted at 5 V/cm^2^ for 60 min. Stained bands were visualized using gel imaging equipment, as depicted in Figure 7.

Each gel well was loaded with a mixture of 3 µL loading dye and 7 µL of extracted genomic DNA or PCR product. Once all wells were loaded, electrophoresis was run at 100 volts (5 V/cm^2^) for 60 min, allowing negatively charged DNA to migrate from the cathode (−) to the anode (+).

Ethidium bromide staining was performed by preparing a solution with 70 µL of 10 mg/mL ethidium bromide in 300 mL distilled water. Gels were soaked in the staining solution for 20–30 min, and then visualized using a gel documentation system at a wavelength of 365 nm. Images of DNA bands were captured and saved using Geneious Prime software version 2025.1 [127].

### 4.8. DNA Sequencing

Sanger sequencing was conducted on the amplified PCR fragments using an ABI3730XL automated DNA sequencer (Macrogen Corporation, Seoul, Republic of Korea) (Appendix B) [128]. Genotypes were identified using BioEdit software by aligning the sequences with the reference from GenBank [116].

BioEdit software is widely used in molecular biology, originally developed as a Windows-based sequence alignment editor. It offers various alignment features, including manual alignment, split-window views, user-defined color schemes, and integration with tools like ClustalW and BLAST. Over time, BioEdit has expanded to include additional functionalities such as plasmid drawing, annotation, and restriction mapping, making it a valuable tool for molecular biologists. Its shareware licensing, efficient modules, and fast performance have contributed to its popularity in molecular research [129]. Primers for FADS1 (rs174547) detection and genotyping were prepared based on reference sequences from the National Center for Biotechnology Information (NCBI). Primer sequences were verified using NCBI bioinformatics tools and are presented in Appendix C.

### 4.9. Sample Size Calculation

Post hoc power analysis was performed using G*Power version 3.1.9.7 [130,131]. With an effect size of 0.25, α-level of 0.05, and 90% power using a t-test family, the required total sample size was 120, with 60 participants in each group.

### 4.10. Statistical Analysis

Statistical analysis was conducted using GraphPad Prism version 9.0 for Windows and SPSS version 24.1. The Anderson–Darling test assessed the normality of variable distributions. Discrete variables were analyzed using the Chi-square test or Fisher’s exact test, as appropriate. Two-sample t-tests were used to compare group means when data followed a normal distribution with no significant outliers. For non-normally distributed data, the Mann–Whitney U test was applied to compare medians between two groups. The Kruskal–Wallis test was used to assess median differences across three groups.

Binary logistic regression analysis was used to calculate odds ratios (ORs) and corresponding 95% confidence intervals for binary outcomes. For multivariate analysis, an unconditional logistic regression model was applied to examine the association between T2DM risk (dependent variable) and genetic polymorphism (independent variables). The analysis followed a hierarchical approach: Model-1 adjusted for sociodemographic variables (sex, age, fish intake, education, BMI, and WHR); Model-2 included lipid profile variables (Apo A, Apo B, total cholesterol, triglycerides, LDL, and HDL) in addition to Model-1; Model-3 further adjusted for hs-CRP; and Model-4 included D5D and D6D levels along with all prior variables.

Allele and genotype frequencies were calculated and presented as counts and percentages. Hardy–Weinberg equilibrium (HWE) was assessed using the SHEsisPlus web tool (http://shesisplus.bio-x.cn/SHEsis.html [Accessed on 20 February 2025]) for two alleles. The χ^2^ goodness-of-fit test compared observed and expected genotype frequencies based on binomial distribution assumptions [132]. A *p*-value < 0.05 was considered statistically significant, where applicable [133,134].

## 5. Conclusions

The C allele of FADS1 (rs174547) was significantly associated with T2DM, being more prevalent in T2DM patients than in controls. Additionally, rs174547 genotype distribution differed significantly between groups, with TT being the most common genotype. D5D enzyme levels were higher in the TC genotype compared to other genotypes, although this was accompanied by lower D5D enzymatic activity; however, the difference was not statistically significant. D6D enzyme levels and activity showed no significant variation across rs174547 genotypes.

## Figures and Tables

**Figure 1 ijms-26-04015-f001:**
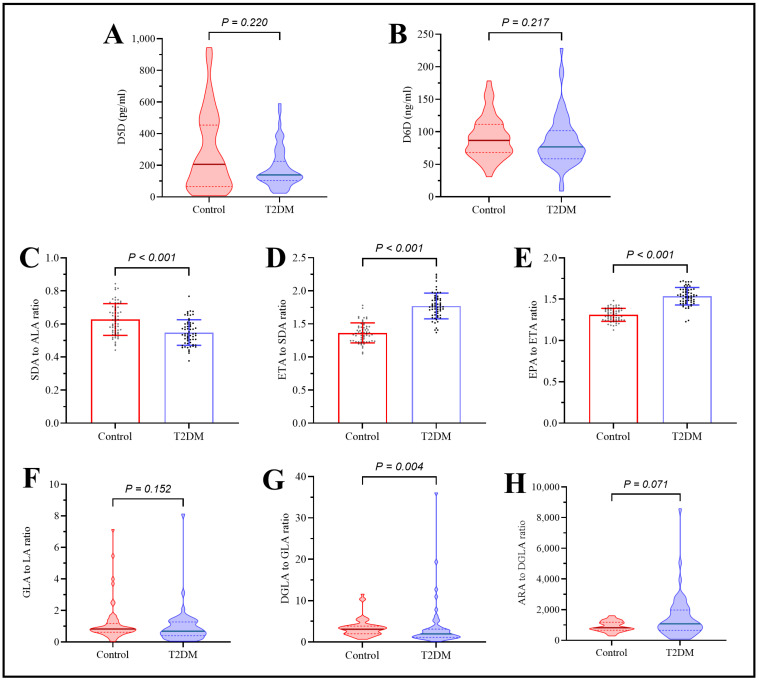
Assessment of plasma desaturase enzyme levels and activities. (**A**) D5D enzyme levels, (**B**) D6D enzyme levels, (**C**) SDA/ALA ratio, (**D**) ETA/SDA ratio, (**E**) EPA/ETA ratio, (**F**) GLA/LA ratio, (**G**) DGLA/GLA ratio, and (**H**) AA/DGLA ratio.

**Figure 2 ijms-26-04015-f002:**
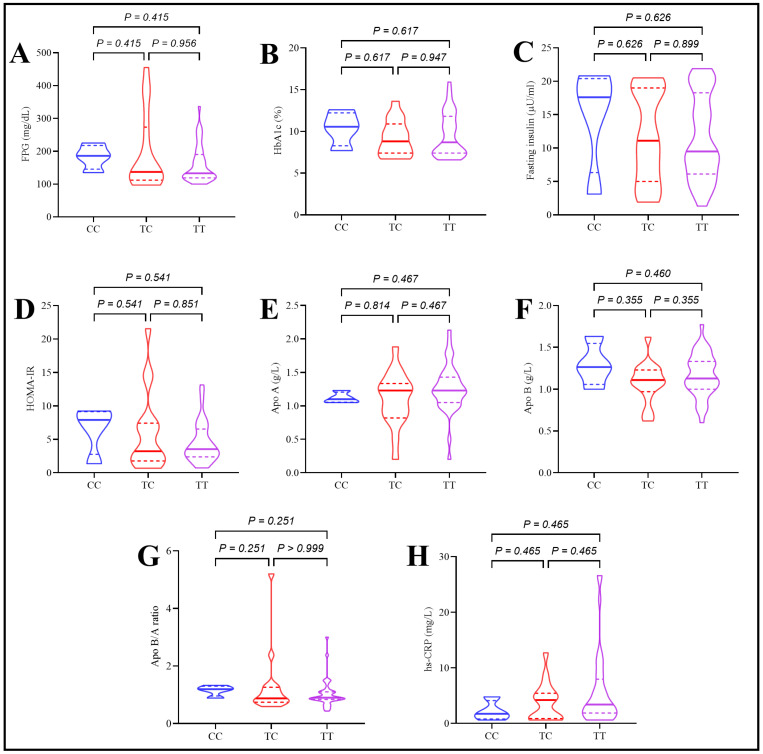
Assessment of glycemic control parameters, apolipoproteins, and hs-CRP in type 2 diabetes patients under the effect of rs174547 polymorphism. (**A**) Fasting plasma glucose, (**B**) HbA1c, (**C**) fasting insulin, (**D**) HOMA-IR, (**E**) serum apoprotein A levels, (**F**) serum apoprotein B levels, (**G**) apoprotein B/A ratio, and (**H**) serum hs-CRP levels.

**Figure 3 ijms-26-04015-f003:**
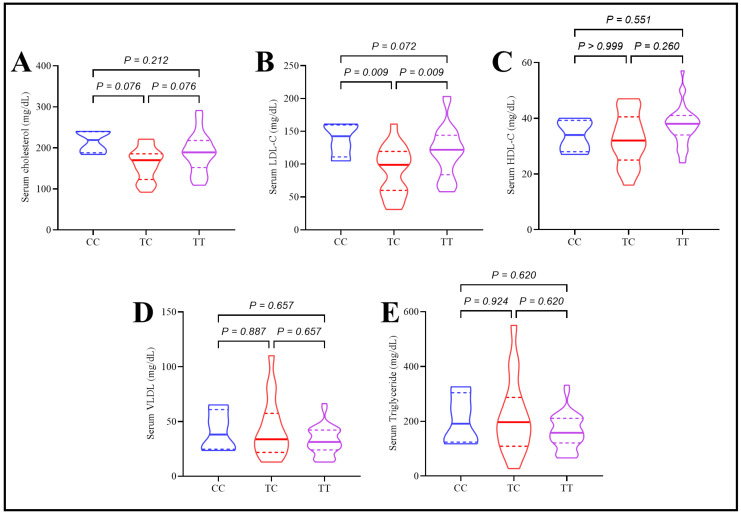
Assessment of lipid panel in type 2 diabetes patients under the effect of rs174547 polymorphism. (**A**) Total serum cholesterol, (**B**) serum LDL-c, (**C**) serum HDL-c, (**D**) serum triglyceride, and (**E**) serum VLDL.

**Figure 4 ijms-26-04015-f004:**
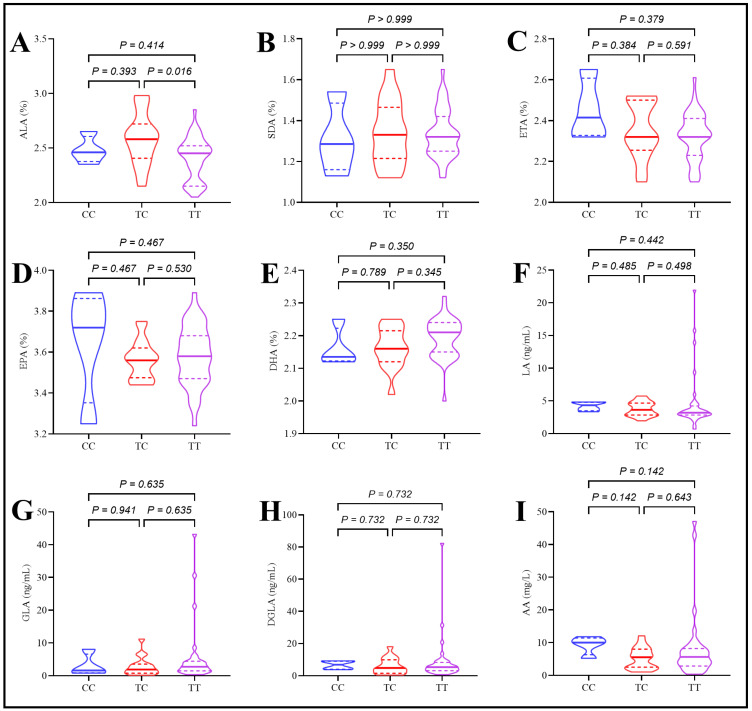
Effect of rs174547 polymorphism on plasma polyunsaturated fatty acids in type 2 diabetes patients (**A**) Serum alpha-linolenic acid levels, (**B**) serum stearidonic acid levels, (**C**) serum eicosatetraenoic acid levels, (**D**) serum eicosapentaenoic acid levels, (**E**) serum docosahexaenoic acid levels, (**F**) serum linolic acid levels, (**G**) serum Gamma-linolenic acid levels, (**H**) serum dihomo-Gamma-linolenic acid levels, (**I**) serum arachidonic acid levels.

**Figure 5 ijms-26-04015-f005:**
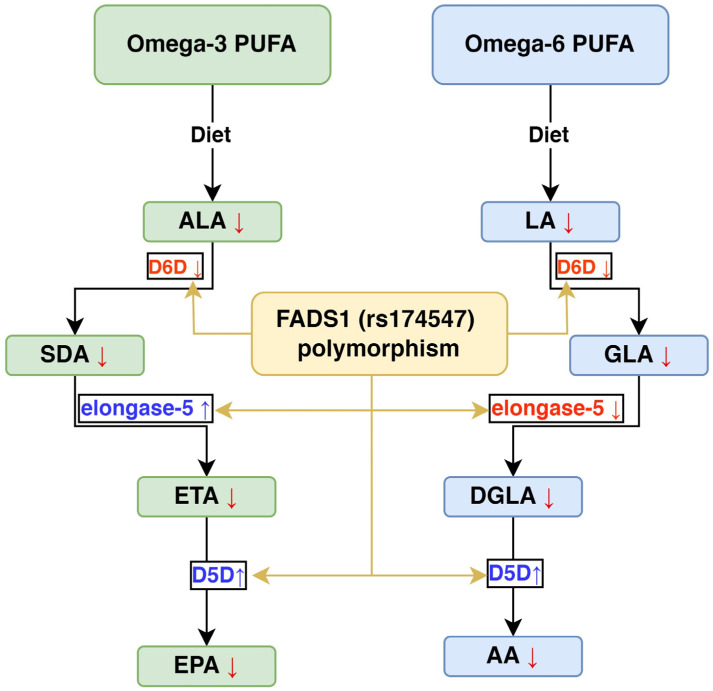
Effect of *FADS1* gene polymorphism (re174547) on PUFA and desaturase enzymes in type 2 diabetes patients. ALA: alpha-linolenic acid, SDA: stearidonic acid, ETA: eicosatetraenoic acid, EPA: eicosapentaenoic acid, DHA: docosahexaenoic acid, LA: linolic acid, GLA: Gamma-linolenic acid, DGLA: dihomo-Gamma-linolenic acid, AA: arachidonic acid, PUFA: polyunsaturated fatty acid.

**Figure 6 ijms-26-04015-f006:**
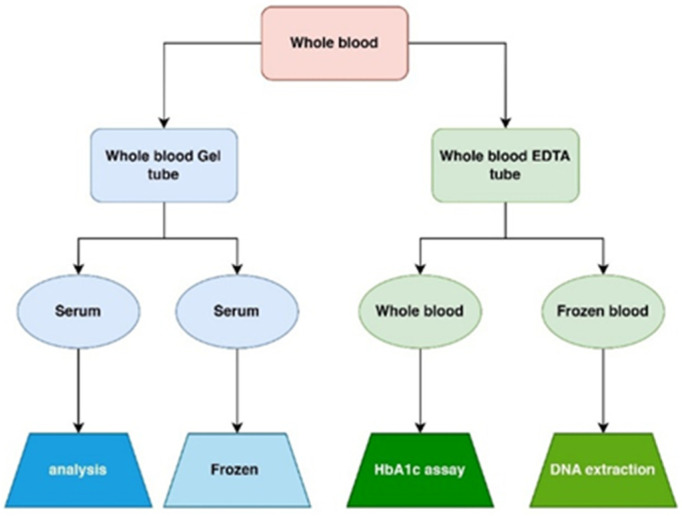
Flow chart of specimen collection.

**Figure 7 ijms-26-04015-f007:**
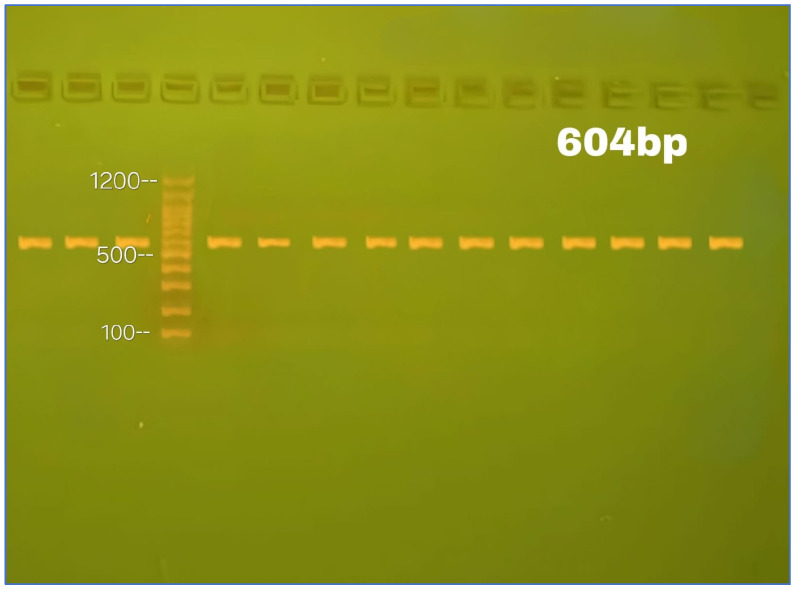
Electrophoresis of the *FADS1* gene PCR reaction outcomes in a 604 bp PCR product size. A DNA marker ladder (100–1200 bp) and a 2% agarose gel were utilized.

**Table 1 ijms-26-04015-t001:** Assessment of sociodemographic variables, glycemic control parameters, lipid-related parameters, polyunsaturated fatty acids, and inflammatory markers.

Variables	Control	T2DM	*p*-Value
Number	60	60	-
Age (years) ^a^	45.6 ± 6.7	47.8 ± 9.9	0.160 ^d^
Sex ^b^			0.264 ^e^
Female	27 (45.0%)	21 (35.0%)	
Male	33 (55.0%)	39 (65.0%)	
BMI (kg/m^2^) ^a^	27.2 ± 2.1	27.0 ± 2.0	0.566 ^d^
WHR ^a^	0.837 ± 0.041	0.841 ± 0.039	0.552 ^d^
Education levels ^b^			0.484 ^e^
Illiterate	6 (10.0%)	11 (18.3%)	
Primary or secondary	51 (85.0%)	47 (78.3%)	
College	3 (5.0%)	2 (3.3%)	
Fish diet ^b^			0.575 ^e^
Once every two weeks	10 (16.7%)	14 (23.3%)	
Once weekly	33 (55.0%)	28 (46.7%)	
Twice weekly	17 (28.3%)	18 (30.0%)	
FPG (mg/dL) ^c^	98 (93–105)	136.5 (119.3–198.8)	<0.001 ^f^
HbA1c (%) ^a^	5.73 ± 0.41	9.50 ± 2.42	<0.001 ^d^
Fasting insulin (µU/mL) ^c^	8.6 (4.52–11.85)	10.21 (5.89–18.96)	0.009 ^f^
HOMA-IR ^c^	2.2 (1.11–2.78)	3.63 (2.34–7.04)	<0.001 ^f^
Apo A (g/L) ^a^	1.28 ± 0.25	1.21 ± 0.36	0.210 ^d^
Apo B (g/L) ^a^	1.15 ± 0.22	1.15 ± 0.25	0.920 ^d^
Apo B/A ratio ^c^	0.87 (0.75–1.11)	0.90 (0.82–1.16)	0.320 ^f^
Cholesterol (mg/DL) ^a^	173.8 ± 31.7	182.1 ± 45.6	0.250 ^d^
LDL-c (mg/dL) ^a^	110.6 ± 24.9	111.7 ± 39.3	0.860 ^d^
HDL-c (mg/dL) ^a^	39.95 ± 6.91	35.72 ± 7.66	0.002 ^d^
Triglyceride (mg/dL) ^c^	108.5 (88.5–142.8)	167.5 (118.8–215.0)	<0.001 ^f^
VLDL (mg/dL) ^c^	21.9 (17.7–21.9)	32.9 (23.7–42.95)	<0.001 ^f^
hs-CRP (mg/L) ^c^	1.78 (0.6–4.27)	3.28 (1.16–5.83)	0.020 ^f^
ω3-PUFA			
ALA (%) ^a^	4.014 ± 0.413	2.46 ± 0.217	<0.001 ^d^
SDA (%) ^a^	2.482 ± 0.216	1.334 ± 0.131	<0.001 ^d^
ETA (%) ^a^	3.353 ± 0.160	2.34 ± 0.126	<0.001 ^d^
EPA (%) ^a^	4.378 ± 0.172	3.581 ± 0.143	<0.001 ^d^
DHA (%) ^a^	2.513 ± 0.102	2.182 ± 0.061	<0.001 ^d^
ω6-PUFA			
LA (ng/mL) ^c^	4.71 (3.19–7.09)	3.41 (2.86–4.49)	0.009 ^f^
GLA (ng/mL) ^c^	4.32 (2.98–5.69)	2.44 (1.24–4.21)	0.001 ^f^
DGLA (ng/mL) ^c^	12.13 (6.94–19.34)	4.91 (3.24–8.79)	<0.001 ^f^
AA (mg/L)	10.41 (5.997–14.71)	5.65 (2.57–8.34)	<0.001 ^f^

^a^ Data presented as mean ± standard deviation, ^b^ as number (%), ^c^ as median (interquartile range), ^d^ independent *t*-test, ^e^ Chi-square test, ^f^ Mann–Whitney U test. BMI: body mass index, WHR: Waist-to-hip ratio, HOMA-IR: Homeostatic Model Assessment for Insulin Resistance, FPG: fasting plasma glucose, HbA1c: glycated hemoglobin, LDL: low-density lipoprotein, HDL: high-density lipoprotein, VLDL: very-low-density lipoprotein, hs-CRP: high-sensitivity C-reactive protein, ALA: alpha-linolenic acid, SDA: stearidonic acid, ETA: eicosatetraenoic acid, EPA: eicosapentaenoic acid, DHA: docosahexaenoic acid, LA: linolic acid, GLA: Gamma-linolenic acid, DGLA: dihomo-Gamma-linolenic acid, AA: arachidonic acid, PUFA: polyunsaturated fatty acid.

**Table 2 ijms-26-04015-t002:** Allele distribution and association between *FADS1* (rs174547) genotype with risk of type 2 diabetes according to genetic models.

Models	Genotype	Control	T2DM	OR (95% CI)	*p*-Value
Co-dominant	CC	2 (3.3%)	4 (6.7%)	2.615 (0.455–15.018)	0.281 ^a^
	TC	7 (11.7%)	17 (28.3%)	3.176 (1.199–8.411)	0.020 ^a^
	TT	51 (85.0%)	39 (65.0%)	1.0	-
Dominant	CC + TC	9 (15.0%)	21 (35.0%)	3.051 (1.259–7.395)	0.014 ^a^
	TT	51 (85.0%)	39 (65.0%)	1.0	-
Recessive	TT + TC	58 (96.7%)	56 (93.3%)	0.483 (0.085–2.741)	0.411 ^a^
	CC	2 (3.3%)	4 (6.7%)	1.0	-
Allele	C	11 (9.17%)	25 (20.83%)	-	0.011 ^b^
	T	109 (90.83%)	95 (79.17%)	-	

^a^ Binary logistic regression analysis, ^b^ Chi-square test. CI: confidence interval, OR: odds ratio.

**Table 3 ijms-26-04015-t003:** Multivariate analysis of the association between *FADS1* (rs174547) genotype to predict type 2 diabetes.

Models	OR (95% CI)	*p*-Value
**Unadjusted ^a^**	3.051 (1.259–7.395)	0.014
**Model-1 ^b^**	3.266 (1.279–8.338)	0.013
**Model-2 ^c^**	3.230 (1.034–10.087)	0.044
**Model-3 ^d^**	3.967 (1.167–13.480)	0.027
**Model-4 ^e^**	3.070 (0.777–12.135)	0.110

^a^ Dominant model (CC + TC vs. TT). ^b^ Adjustment for sociodemographic (sex, age, fish diet, education levels, BMI, and WHR). ^c^ Adjustment for lipid profile: apo A, apo B, total cholesterol, triglycerides, LDL, and HDL (in addition to Model-1 parameters). ^d^ Model-3: adjustment for hs-CRP (in addition to Model-2 parameters). ^e^ Model-4: adjustment for D5D and D6D (in addition to Model-3 parameters).

**Table 4 ijms-26-04015-t004:** Effect of rs174547 polymorphism on desaturase enzymes in type 2 diabetes patients.

Variables ^a^	CC	TC	TT	*p*-Value ^b^
Number	4	17	39	-
D5D (pg/mL)	164.5 (118.6–262.5)	177.7 (113.6–307.3)	134.9 (85.1–213.8)	0.243
EPA/ETA ratio (D5D)	1.5 (1.3–1.7)	1.5 (1.5–1.6)	1.5 (1.5–1.6)	0.480
AA/DGLA ratio (D5D)	1625.7 (768.4–2561.5)	825.3 (494.7–2533.5)	1075.9 (688.7–1809.6)	0.701
D6D (ng/mL)	79.9 (58.9–97.8)	89.6 (55.6–104.7)	76.7 (60.4–102.2)	0.986
SDA/ALA ratio (D6D)	0.5 (0.5–0.6)	0.5 (0.5–0.6)	0.6 (0.5–0.6)	0.188
GLA/LA ratio (D6D)	0.5 (0.2–1.4)	0.6 (0.2–1.3)	0.8 (0.5–1.3)	0.500
ETA/SDA ratio (elongase-5)	1.8 (1.7–2.1)	1.7 (1.6–1.9)	1.8 (1.6–1.9)	0.562
DGLA/GLA ratio (elongase-5)	3.3 (1.3–9.5)	1.7 (1.1–3.1)	2.1 (1.1–3.0)	0.692

^a^ Data presented as median (IQR), ^b^ Kruskal–Wallis test. D5D: delta-5 desaturase, D6D: delta-6 desaturase, ALA: alpha-linolenic acid, SDA: stearidonic acid, ETA: eicosatetraenoic acid, EPA: eicosapentaenoic acid, LA: linolic acid, GLA: Gamma-linolenic acid, DGLA: dihomo-Gamma-linolenic acid, AA: arachidonic acid.

## Data Availability

The data presented in this study are openly available in Zenodo at https://doi.org/10.5281/zenodo.15021480, reference number 15021480.

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
