# Peer review of "Influence of Fatty Acid Desaturase Enzyme-1 Gene (FADS-1) Polymorphism on Serum Polyunsaturated Fatty Acids Levels, Desaturase Enzymes, Lipid Profile, and Glycemic Control Parameters in Newly Diagnosed Diabetic Mellitus Patients"

_ijms, 2025, doi:10.3390/ijms26094015_

Round 1
Reviewer 1 Report
Comments and Suggestions for Authors
I would suggest minor revisions. Polymorphisms are of interest, but would be enhanced by a much more expansive discussion of why these polymorphisms would correlate with type 2 Diabetes. They mention (lines 50-52) a general "capacity of various dietary fatty acids toinitiate disctinct intracellular signalig pathways" but these desaturases are presumably involved with endogenous faty synthesis such as fatty acids made from glucose (explained later in 66-73). The authors later state that arachidonic acid and eicosapenaenoic acid serve as precursors to bioactive lipid mediators (lines 120-123) but the discussion is vague and non-specific. The work needs more rationale as to why D5D and D6D would correlate with type 2 diabetes. The authors could have talked about PUFA and membrane fluidity affecting insulin binding; the role of PUFAs as substrates in signaling pathways, or the role of PUFAs in transcription complexes and effects on transcription factors such as SREBP1, with details. The paper would also be enhanced by a figure describing the chemistry of the enzymes and why these polymorphisms would affect the activity. The other thing that I struggled to understand was this: if the polymorphisms are associated with type 2 diabetes, then why ask about things like a fish diet? Does diet factor in to these measurements of insulin, HbA1c, cholesterol, LDL, HDL, etc.? Again, rationale is missing for what was measured.
Comments on the Quality of English LanguageA few minor errors, mostly in the abstract.
Author Response
Thank you very much for taking the time to review this manuscript. Please find the detailed responses below and the corrections highlighted in the re-submitted files.
1/ Polymorphisms are of interest, but would be enhanced by a much more expansive discussion of why these polymorphisms would correlate with type 2 Diabetes.
Answer
Thank you for pointing this out; we agree with this reviewer's comment.
Regarding the mechanism of association between polymorphism and risk of T2DM,
One possible explanation is that the C alleles of FADS1 rs174547 may hinder desaturase activity, resulting in the diminished production of n-6 and n-3 long-chain PUFAs, specifically 20:4n-6 and 20:5n-3 [1].
2/ They mention (lines 50-52) a general "capacity of various dietary fatty acids to initiate distinct intracellular signaling pathways," but these desaturases are presumably involved with endogenous fatty synthesis, such as fatty acids made from glucose (explained later in 66-73). The authors later state that arachidonic acid and eicosapentaenoic acid serve as precursors to bioactive lipid mediators (lines 120-123), but the discussion is vague and non-specific.
Answer
Thank you for pointing this out; we want to explain this section in the introduction.
Polyunsaturated fatty acids (PUFAs), including omega-3 and omega-6 fatty acids, are essential fats that the human body cannot produce on its own. Therefore, they must be obtained through dietary sources. The first section highlights their importance after entering the human body and their biological functions.
The section is directed to explain how the human body synthesizes the rest of FFA from their dietary sources.
The third section is focused on the activity of FFA products like arachidonic acid and their possible pathological/ biological activity.
3/ The work needs more rationale as to why D5D and D6D would correlate with type 2 diabetes.
Answer
Thank you for pointing this out; we agree with this reviewer's comment. We add the rationale for the correlation between D5D and D6D with the risk of T2DM as follows:
In the present study, D5D and D6D levels were lower in T2DM patients; in comparison, the enzymatic activity of D5D was higher in T2DM patients, while D6D enzyme activity was higher in T2DM patients. A Mendelian randomization study indicated that the genetically-determined low D5D activity tended to predict higher diabetes risk [2]. The molecular processes connecting D5D and D6D to diabetes risk remain poorly elucidated. Cross-sectionally, modified D5D and D6D activity were associated with insulin resistance [3,4], and lifestyle-induced alterations in D5D and D6D activity were correlated with variations in insulin sensitivity [5]. The rs174550 genotype in FADS1, which is in complete linkage disequilibrium with rs174546, was correlated with early insulin secretion but not with insulin sensitivity [6]. A separate study found no consistent correlation between the rs174550 genotype and β cell function [7]. The impact of desaturase activity on diabetes risk is probably mediated by alterations in fatty acid composition. The fatty acid content of cell membranes may influence cellular function, including insulin signaling and receptor binding affinities. Moreover, LC-PUFA may function as ligands for transcription factors, including sterol regulatory element-binding protein 1 and peroxisome proliferator-activated receptors [8-10].
4/ The authors could have talked about PUFA and membrane fluidity affecting insulin binding; the role of PUFAs as substrates in signaling pathways, or the role of PUFAs in transcription complexes and effects on transcription factors such as SREBP1, with details.
Answer
Thank you for pointing this out; we agree with this reviewer's comment. We added the following sections to the discussion:
PUFAs are essential constituents of all cellular membranes. They are incorporated within phospholipids and other intricate lipids, where they are esterified to glycerol or other polyols. These lipids frequently possess hydrophilic molecules that enhance their structural and functional roles in cellular membranes [11-13]. The incorporation of saturated fats into cell membranes results in a densely packed and less flexible lipid bilayer, attributable to their straight-chain structure. Conversely, PUFAs establish a more fluid and dynamic membrane environment due to their numerous double bonds and bent structure. The variation in lipid content directly influences membrane fluidity, which is essential for sustaining normal cellular processes, such as receptor mobility, intracellular signaling, and effective nutrition and ion transport. A stiff membrane structure caused by saturated fat accumulation can obstruct the normal functioning of insulin receptors by diminishing their mobility and affecting their interaction with signaling proteins. This molecular imbalance impairs insulin signaling pathways, leading to insulin resistance, a significant contributor to the onset of T2DM [14,15].
Furthermore, Omega-3 fatty acids are crucial for preserving insulin sensitivity and modulating glucose metabolism. They augment insulin efficacy in target tissues such as muscle and adipose tissue by amplifying signaling pathways linked to insulin receptors. Omega-3 fatty acids stimulate the expression of glucose transporter type 4 (GLUT4) via methods involving the peroxisome proliferator-activated receptor gamma (PPAR-γ), a crucial transcription factor that improves insulin sensitivity. Low Omega-3 levels, particularly in a diet rich in Omega-6 fatty acids, disturb the fatty acid composition balance in cell membranes. This imbalance may result in excessive production of pro-inflammatory mediators from Omega-6 fatty acids, exacerbating systemic inflammation and insulin resistance. Chronic inflammation disrupts insulin signaling, diminishes glucose absorption, and elevates the risk of type 2 diabetes development [16].
Studies indicate that omega-3 fatty acids regulate triglyceride levels via modulating the activity of many nuclear receptors, including sterol regulatory element-binding protein (SREBP), liver X receptor-alpha (LXRα), retinoid X receptor-alpha (RXRα), farnesoid X receptor (FXR), and peroxisome proliferator-activated receptors (PPARs) [17]. SREBPs are transcription factors that regulate the expression of genes associated with lipid synthesis and are involved in various physiological and pathological processes [18]. Omega-3 fatty acids reduce hepatic fat synthesis by suppressing SREBP-1c activity. It is thought to decrease the synthesis of VLDL, fatty acids, and triglyceride-producing enzymes [19]. Omega-3 fatty acids exert an inhibitory effect on SREBP-1c expression by obstructing the binding of the LXR/RXR heterodimer to the LXR response elements (LXREs) inside the SREBP-1c promoter, a crucial step for SREBP-1c expression [20]. The accumulation of lipids in non-adipose tissues is frequently linked to Type 2 diabetes and its related consequences. Increased expression of the lipogenic transcription factor, SREBP-1c, has been observed in the islets and liver of diabetic rats [21]; thus, lower levels of omega-3 FFA will lead to higher activity of SREBP and higher risk of T2DM.
The significance of endoplasmic reticulum (ER) stress in terms of insulin resistance has been emphasized. An imbalance in fatty acid content can trigger endoplasmic reticulum stress, triggering the unfolded protein response (UPR) and therefore disrupting insulin signaling. The complex interaction among fatty acid composition, inflammation, and cellular stress highlights the crucial importance of sustaining a balanced intake of PUFAs to prevent and manage insulin resistance and type 2 diabetes. Omega-3 fatty acids improve insulin sensitivity via PPAR-gamma activation and anti-inflammatory processes. Conversely, high consumption of Omega-6 fatty acids intensifies insulin resistance through inflammatory mechanisms, dysregulated adipokine release, and the induction of endoplasmic reticulum (ER) stress [22,23].
5/ The paper would also be enhanced by a figure describing the chemistry of the enzymes and why these polymorphisms would affect the activity.
Answer
Thank you for pointing this out; we agree with this reviewer's comment. We added the following Figure:
Figure 5: Effect of FADS1 gene polymorphism (re174547) on PUFA and desaturase enzymes in type 2 diabetes patients
6/ The other thing that I struggled to understand was this: if the polymorphisms are associated with type 2 diabetes, then why ask about things like a fish diet? Does diet factor into these measurements of insulin, HbA1c, cholesterol, LDL, HDL, etc.?
Answer
Thank you for pointing this out; we agree with this reviewer's comment. We examined the fish diet at baseline to establish no differences between both groups, thus excluding it as a possible confounder.
7/ Again, the rationale is missing for what was measured.
Answer
Thank you for pointing this out; we measured all parameters that could give us ideas about the relationship between T2DM and lipid parameters and free fatty acid synthesis, in addition to examining the sociodemographic and clinical characteristics of the participants.

Reviewer 2 Report
Comments and Suggestions for Authors
Hi Authors,
Congratulations for the article, and appreciate your contribution to this field, and I have the following comments.
- Is it possible to give more explanation on the biological significance of FADS1 (rs174547) gene and possible to mention factors like dietary influences and lifestyle factors effects on the lipid metabolism and enzymatic activity (positively or negatively correlated). This would strengthen the foundation of the work.
- A mechanistic link between gene and lipid metabolism, maybe like a figure would be useful.
- Possibility for the findings in this study to be used for personalized medicine (perspective)? It is possible to give genetic insights contribution to individualized treatments for diabetes mellitus discussed in the article.
- In the tables, include p -values to give clear idea on statistical significance.
Thank you very much.
Author Response
Thank you very much for taking the time to review this manuscript. Please find the detailed responses below and the corrections highlighted in the re-submitted files
Hi Authors,
Congratulations for the article, and appreciate your contribution to this field, and I have the following comments.
1/ Is it possible to give more explanation on the biological significance of FADS1 (rs174547) gene and possible to mention factors like dietary influences and lifestyle factors effects on the lipid metabolism and enzymatic activity (positively or negatively correlated). This would strengthen the foundation of the work.
Answer
Thank you for pointing this out; we agree with this reviewer's comment. We added the following to the discussion:
The interplay between genetic and non-genetic factors complicates the association between ω-3 PUFA consumption and T2DM. Polymorphisms in the FADS gene have been shown to influence the function of ω-3 PUFA in T2DM. The endogenous metabolism of ω-3 PUFA is facilitated by D5D and D6D, which are encoded by the FADS1 and FADS2 genes, respectively. Genetic diversity in FADS genes correlates with diminished expression and activity of D5D and D6D, impacting the concentration of ω-3 PUFA [40].
In a pharmacokinetic study of 161 T2DM patients, a dose-response relationship between ω-3 PUFA plasma level and HbA1c was established. The authors utilized nonlinear mixed-effect analysis, indicating the complexity of this relationship [22]. This study found that HDL is a determinant variable that influenced this relationship, in which individuals with reduced HDL, the clearance of ω-3 PUFA was faster, and the effectiveness in lowering HbA1c was consequently lowered; this may elucidate the critical instance of HDL levels in enhancing HbA1c [22]. A cross-sectional investigation identified a U-shaped relationship between HDL-C and HbA1c in individuals with type 2 diabetes aged 60 years and above, with an HDL-C inflection point of 60 mg/dL [41]. Dietary PUFA, notably ω-3 PUFA, promotes the transport of HDL-containing apoE via several protective mechanisms that involve apoE; the apoE facilitates multiple mechanisms that characterize reverse cholesterol transport via HDL, including the secretion of active HDL subspecies, cholesterol efflux to HDL from macrophages implicated in atherogenesis, the enlargement of HDL size with cholesterol ester, and expedited clearance from the bloodstream; thus reducing the risk of T2DM. ApoC3 negates the advantageous effects of apoE on reverse cholesterol transport, potentially explaining the correlation between HDL-containing apoC3 and T2DM, along with other disorders [42]. In the current study, HDL levels were significantly lower in T2DM compared to healthy participants, which further validated Wang et al. study [22], indicating the complex interplay between PUFA, HDL, and the risk of T2DM. In the current study, polymorphism in rs174547 did not influence HDL-C, which indicates that the effect of HDL-C on the risk of T2DM was not influenced by FADS1 gene polymorphism.
PUFAs are essential constituents of all cellular membranes. They are incorporated within phospholipids and other intricate lipids, where they are esterified to glycerol or other polyols. These lipids frequently possess hydrophilic molecules that enhance their structural and functional roles in cellular membranes [43-45]. Incorporating saturated fats into cell membranes results in a densely packed and less flexible lipid bilayer, attributable to their straight-chain structure. Conversely, PUFAs establish a more fluid and dynamic membrane environment due to their numerous double bonds and bent structure. The variation in lipid content directly influences membrane fluidity, essential for sustaining normal cellular processes, such as receptor mobility, intracellular signaling, and effective nutrition and ion transport. A stiff membrane structure caused by saturated fat accumulation can obstruct the normal functioning of insulin receptors by diminishing their mobility and affecting their interaction with signaling proteins. This molecular imbalance impairs insulin signaling pathways, leading to insulin resistance, a significant contributor to the onset of T2DM [46,47].
Furthermore, Omega-3 fatty acids are crucial for preserving insulin sensitivity and modulating glucose metabolism. They augment insulin efficacy in target tissues such as muscle and adipose tissue by amplifying signaling pathways linked to insulin receptors. Omega-3 fatty acids stimulate the expression of glucose transporter type 4 (GLUT4) via methods involving the peroxisome proliferator-activated receptor gamma (PPAR-γ), a crucial transcription factor that improves insulin sensitivity. Low Omega-3 levels, particularly in a diet rich in Omega-6 fatty acids, disturb the fatty acid composition balance in cell membranes. This imbalance may produce excessive pro-inflammatory mediators from Omega-6 fatty acids, exacerbating systemic inflammation and insulin resistance. Chronic inflammation disrupts insulin signaling, diminishes glucose absorption, and elevates the risk of type 2 diabetes development [48].
Studies indicate that omega-3 fatty acids regulate triglyceride levels via modulating the activity of many nuclear receptors, including sterol regulatory element-binding protein (SREBP), liver X receptor-alpha (LXRα), retinoid X receptor-alpha (RXRα), farnesoid X receptor (FXR), and peroxisome proliferator-activated receptors (PPARs) [49]. SREBPs are transcription factors that regulate the expression of genes associated with lipid synthesis and are involved in various physiological and pathological processes [50]. Omega-3 fatty acids reduce hepatic fat synthesis by suppressing SREBP-1c activity. It is thought to decrease the synthesis of VLDL, fatty acids, and triglyceride-producing enzymes [51]. Omega-3 fatty acids exert an inhibitory effect on SREBP-1c expression by obstructing the binding of the LXR/RXR heterodimer to the LXR response elements (LXREs) inside the SREBP-1c promoter, a crucial step for SREBP-1c expression [52]. The accumulation of lipids in non-adipose tissues is frequently linked to Type 2 diabetes and its related consequences. Increased expression of the lipogenic transcription factor, SREBP-1c, has been observed in the islets and liver of diabetic rats [53]; thus, lower levels of omega-3 FFA will lead to higher activity of SREBP and higher risk of T2DM.
Endoplasmic reticulum (ER) stress's significance in insulin resistance has been emphasized. An imbalance in fatty acid content can trigger endoplasmic reticulum stress, triggering the unfolded protein response (UPR) and disrupting insulin signaling. The complex interaction among fatty acid composition, inflammation, and cellular stress highlights the crucial importance of sustaining a balanced intake of PUFAs to prevent and manage insulin resistance and type 2 diabetes. Omega-3 fatty acids improve insulin sensitivity via PPAR-gamma activation and anti-inflammatory processes. Conversely, high consumption of Omega-6 fatty acids intensifies insulin resistance through inflammatory mechanisms, dysregulated adipokine release, and the induction of endoplasmic reticulum (ER) stress [54,55].
2/ A mechanistic link between gene and lipid metabolism, maybe like a figure, would be useful.
Answer
Thank you for pointing this out; we agree with this reviewer's comment. We added the following Figure:
Figure 5: Effect of FADS1 gene polymorphism (re174547) on PUFA and desaturase enzymes in type 2 diabetes patients
3/ Possibility for the findings in this study to be used for personalized medicine (perspective)? It is possible to give genetic insights contribution to individualized treatments for diabetes mellitus discussed in the article.
Answer
Thank you for pointing this out; we agree with this reviewer's comment. We added the following to the discussion:
The current study's findings can be translated to offer a deep understanding of the optimal supplementation of omega-3. Diabetic patients showed lower PUFA in their serum, which was correlated with low D5D and D5D levels and activity. Furthermore, patients with TT-polymorphism of rs174547 showed lower levels and enzymatic activity of D5D and D6D, which indicates they even require higher doses of supplementary LA and ALA.
4/ In the tables, include p -values to give clear idea on statistical significance.
Answer
Thank you for pointing this out; all Table now have p-values.
Thank you very much.

Reviewer 3 Report
Comments and Suggestions for Authors
The manuscript investigates the impact of FADS-1 gene polymorphisms on metabolic parameters in newly diagnosed diabetes mellitus patients. The study holds potential but requires major revisions:
1. Introduction: The introduction is too lengthy and could be more concise. Streamlining background information to focus on the study's objectives and research question would enhance clarity.
2. Materials and Methods:
- The methods section needs better clarity, particularly regarding DNA extraction, SNP selection, and genotyping methods. More detail is required to ensure reproducibility.
- Sample size determination lacks explanation. The manuscript should provide details on how the sample size was calculated, including effect size, power analysis, and the rationale for participant numbers to justify the study's statistical power.
3. Structure: The manuscript would benefit from improved organization. The Materials and Methods section should be clearly divided into subsections (e.g., "Sample Selection," "Genotyping," "Statistical Analysis") for better readability and flow.
In summary, the manuscript needs to be more concise and better organized, with added details on sample size calculation and methodology to strengthen its rigor before reconsideration for publication.
Author Response
Thank you very much for taking the time to review this manuscript. Please find the detailed responses below and the corrections highlighted in the re-submitted files.
The manuscript investigates the impact of FADS-1 gene polymorphisms on metabolic parameters in newly diagnosed diabetes mellitus patients. The study holds potential but requires major revisions:
1/ Introduction: The introduction is too lengthy and could be more concise. Streamlining background information to focus on the study's objectives and research question would enhance clarity.
Answer
Thank you for pointing this out; we agree with this reviewer's comment. We rearranged the introduction to the following: “ Diabetes and its prevalence, Polyunsaturated fatty acids, definition, synthesis, biological activity, desaturases enzyme, activity and genetic location, rationale and gap of knowledge of the study.”
2/ Materials and Methods: The methods section needs better clarity, particularly regarding DNA extraction, SNP selection, and genotyping methods. More detail is required to ensure reproducibility.
Thank you for pointing this out; we agree with this reviewer's comment. We adhered to STROBE guidelines for reporting observational studies. Additionally, several supplemental appendices describe the details regarding genetic analysis.
|
Section/Topic |
Recommendation |
|
Study design |
Present key elements of study design early in the paper |
|
Setting |
Describe the setting, locations, and relevant dates, including periods of recruitment, exposure, follow-up, and data collection |
|
Participants |
(a) Case-control study—Give the eligibility criteria, and the sources and methods of case ascertainment and control selection. Give the rationale for the choice of cases and controls |
|
(b) Case-control study—For matched studies, give matching criteria and the number of controls per case |
|
|
Variables |
Clearly define all outcomes, exposures, predictors, potential confounders, and effect modifiers. Give diagnostic criteria, if applicable |
|
Data sources/measurement |
For each variable of interest, give sources of data and details of methods of assessment (measurement). Describe comparability of assessment methods if there is more than one group |
|
Bias |
Describe any efforts to address potential sources of bias |
|
Study size |
Explain how the study size was arrived at |
|
Quantitative variables |
Explain how quantitative variables were handled in the analyses. If applicable, describe which groupings were chosen and why |
|
Statistical methods |
(a) Describe all statistical methods, including those used to control for confounding |
|
(b) Describe any methods used to examine subgroups and interactions |
|
|
(c) Explain how missing data were addressed |
|
|
(d) Cohort study—If applicable, explain how loss to follow-up was addressed Case-control study—If applicable, explain how matching of cases and controls was addressed Cross-sectional study—If applicable, describe analytical methods taking account of sampling strategy |
|
|
(e) Describe any sensitivity analyses |
2.7 Genomic Examination
For deoxynucleic acid extraction, the whole blood frozen collected in EDTA tubes at -40℃ is the sample type utilized in DNA extraction. The whole blood genomic DNA extraction requires cell lysis, protein removal, DNA binding, washing, and elution; this thorough methodology, according to the manufacturer's instructions, EasyPure® Blood Genomic DNA Kit (TransGen, biotech. EE121-02). It offers a straightforward and efficient method for isolating high-quality genomic DNA from 5 – 250 µl of frozen blood [24,25].
20 µl of proteinase K, 500 µl of BB3, and 20 µl of RNase were introduced into a microcentrifuge tube, then vortexing the mixture for 15 seconds; the mixture was incubated at an ambient temperature for 10 minutes. The mixture was centrifuged, then the lysate was added to a spin column and centrifuged at 12,000 rpm for 1 minute, and the flow-through solution was discarded. 500 µl of buffer solution with ethanol (95 – 100%) was added to the spin column and centrifuged at 12,000 rpm for 30 seconds, and the flow-through solution was discarded. Repeat the washing process twice to remove any residual contaminants, and the final DNA preparation is as clean as possible. The spin column is placed into a collection tube, centrifuged at 12,000 rpm for 2 minutes, then dried at room temperature for several minutes. The spin column was placed into a 1.5 ml microcentrifuge tube, and 50-200 µl of elution buffer (prewarmed to 60°C for higher yield) or distilled water (pH > 7.0) was added to the center of the column. It was incubated at room temperature for 1 minute, then centrifuged at 12,000 rpm for 1 minute to elute the genomic DNA. The presence and integrity of the extracted genomic DNA were evaluated on an agarose gel stained with ethidium bromide nucleic acid staining solution for one hour by applying a 100-volt current; this was accomplished by running 10 μL of each sample of the extracted genomic DNA with the solution. Then, the DNA samples were examined using an ultraviolet transilluminator. The electrophoresis of extracted genomic DNA was performed on 1% agarose [26].
The Nanodrop equipment was utilized to assess the purity and concentration of the collected genomic DNA. Light absorbance was measured for each sample at 260 and 280 nm wavelengths. Light absorption at 260 nm is ascribed to DNA, whereas protein contaminants absorb light at 280 nm. The DNA purity may be evaluated using the A260/A280 ratio. A ratio of A260/A280 between 1.7 and 2 signifies good purity of a DNA sample, making it suitable for PCR [27-29]. Polymerase Chain Reaction (PCR) was utilized for isolated genomic DNA amplification.
The primers were designed using Primer3plus [30,31] and double-checked by the University Code of Student Conduct (UCSC) programs [32] and with their reference sequences in the National Center for Biotechnology Information (NCBI) database, as illustrated in Appendix 1. FADS1 (rs174547) primers (F: AGGAGTTGGCTTGGGAAAGT; R: TCTCTGCTCCCACCTGTACC), primer size: 20 bp, product size: 604 bp. For each assay in this study, the required primers were prepared as follows: After dissolving the lyophilized sample in nuclease-free water according to the manufacturer's instructions (Alpha DNA Ltd., Canada), a stock solution with a concentration of 100 μM was prepared and stored at-20°C. Diluting 10 μL of each primer stock solution in 90 μL of nuclease-free water yielded a working solution with a concentration of 10 μM, which was maintained at (-20°C) until use.
A partial sequence was chosen for this study to evaluate the FADS1 gene polymorphism association with T2DM in Iraqi patients [25]. To start the PCR, the reaction was tuned by testing four annealing temperatures: 56, 58, 60, and 62℃. The annealing temperature of 58℃ was the optimum for producing clear and sharp bands in agarose gel; hence, it was used in the current study [33]. This protocol employs 2xEasyTaq® PCR SuperMix. All PCR reactions were carried out in a 25 μl final volume according to the manufacturer's instructions.
The extracted DNA and amplified PCR fragments were separated on an agarose gel and then seen under UV light after ethidium bromide staining [33]: agarose was made at a concentration of (1% (w/v) for DNA extraction and 2% (w/v) for detecting PCR products). One gram of agarose was dissolved in 100 ml of 1X TBE (0.89M Tris, 0.89 M Boric Acid, 2 mM EDTA) buffer (pH=8) and then melted by heating with stirring. After leaving the agarose to cool at 60°C, ethidium bromide was added at a final concentration of 0.5 μg/ml. The agarose was poured into the tray, and a comb was inserted towards one side of the gel. The gel was kept firm until it became opaque, and the comb was carefully removed. After placing the gel tray horizontally in the electrophoresis tank, TBE (1X) buffer was poured into the gel tank until the tray was completely covered (3-5 mm over the surface of the gel). Each wells was filled with 10 μl of extracted DNA or PCR product and 5 μl of DNA ladder (100bp) that acted as a marker throughout the electrophoresis procedure. After running the electrophoresis at 5 v/cm2 for 60 minutes, the Ethidium bromide-stained bands in the gel were observed using the Gel imaging equipment, as illustrated in Figure 7.
The gel's wells were loaded with a mixture of a combination (3 µl of loading dye and 7 µl of extracted genomic DNA (or product PCR)). Following the loading of all wells, the electrical power was turned on for 60 minutes at 100 volts (5 V/cm2); this caused DNA with a negative charge to migrate from the cathode (-) to the anode (+) poles.
After staining electrophoresis gels with ethidium bromide, which was made by adding 70 µl of the 10 mg/ml ethidium bromide to 300 ml of distilled water, the gel was stained by soaking in the solution for 20-30 minutes. Then, the gel was placed into the gel documentation system to view the DNA bands at a 365 nm wavelength. Geneious prime software was utilized to save the photos captured by the device on the computer [34].
2.8 DNA Sequencing
Sanger sequencing was performed on the amplified PCR fragments using an ABI3730XL automated DNA sequencer (Macrogen Corporation, Korea) (Appendix 2) [35]. Bioedit software showed the genotype after aligning with a reference sequence in the Gene Bank [25].
BioEdit is a software that is widely used in molecular biology research. It was initially designed as a Windows-only biological sequence alignment editor. It has various sequence alignment capabilities, such as easy hand alignment, split window view, user-specified color, information-based shading, and auto interaction with other applications like Clustal W and Blast. However, it has been greatly improved in recent years to merge many additional features. It functions as a valuable molecular tool for molecular biologists, and it includes many forms of hand alignment, plasmid drawing and annotation, restriction mapping, and much more. With its versatile molecular biology tools, it has become one of the most extensively used programs in molecular biology. Its shareware licensing, efficient, up-to-date modules, and speedy ability to provide findings make it one of the most popular applications among molecular biologists today [36].
Detecting primers designed for FADS1 (rs174547) gene and genotyping were prepared. The primer sequences were designed following their reference sequences (rs) in the NCBI (National Center for Biotechnology Information) database. NCBI's bioinformatics programs and Appendix 3 matched the genotyping primer sequences.
3/ The Sample size determination lacks explanation. The manuscript should provide details on how the sample size was calculated, including effect size, power analysis, and the rationale for participant numbers to justify the study's statistical power.
Answer
Thank you for pointing this out; we agree with this reviewer's comment. Below is the sample size calculation:
- It was determined using the G*Power version (3.1.9.7) [116,117], based on post-hoc power calculation
- The effect size was 0.25
- α-level 0.05
- 90% power
- T-family test
- The total sample size was 120 (60 in each group).
Based on power analysis, 60 participants in each group were to provide 90% at an effect size of 0.25. next is the calculation from the program:
t tests - Correlation: Point biserial model
Analysis: A priori: Compute required sample size
Input: Tail(s) = Two
Effect size |ρ| = 0.25
α err prob = 0.05
Power (1-β err prob) = 0.9
Output: Noncentrality parameter δ = 3.2659863
Critical t = 1.9750921
Df = 158
Total sample size = 160
Actual power = 0.9007972
4/ Structure: The manuscript would benefit from improved organization. The Materials and Methods section should be clearly divided into subsections (e.g., "Sample Selection," "Genotyping," "Statistical Analysis") for better readability and flow. In summary, the manuscript needs to be more concise and better organized, with added details on sample size calculation and methodology to strengthen its rigor before reconsideration for publication.
Answer
Thank you for pointing this out; we agree with this reviewer's comment. We adhered to STROBE guidelines for reporting observational studies.

Round 2
Reviewer 3 Report
Comments and Suggestions for Authors
The revised manuscript still uses overly complex phrasing in several sections. The writing lacks precision and clarity in key parts, making it harder to follow the methodology and the study's objectives. More straightforward language and clearer sentence structure are needed throughout the manuscript.
Author Response
Comment 1: The revised manuscript still uses overly complex phrasing in several sections. The writing lacks precision and clarity in key parts, making it harder to follow the methodology and the study's objectives. More straightforward language and clearer sentence structure are needed throughout the manuscript.
Response 1: Thank you for pointing this out. We agree with the reviewer; we improved all the manuscript elements, including the study's flow, structure, language, and clarity. We utilized the services of a professional editor (certificate included).

Round 3
Reviewer 3 Report
Comments and Suggestions for Authors
As previously noted, the manuscript format should not be altered. Please ensure that the Results section precedes the Methods, as per the original structure. Consistent adherence to the prescribed format is essential for clarity and alignment with the journal’s requirements.
Author Response
Comment 1: As previously noted, the manuscript format should not be altered. Please ensure that the Results section precedes the Methods, as per the original structure. Consistent adherence to the prescribed format is essential for clarity and alignment with the journal’s requirements.
Response 1: Thank you for pointing this out. We agree with the reviewer; we double-checked the contents of the manuscript and their adherence to the journal's instructions (according to the template). The current manuscript contains the following sections in the following order:
Abstract
1 Introduction
2 Results
3 Discussion
4 Methods
5 Conclusion
In addition, the reviewer's previous comments were addressed by improving all the manuscript elements, including the study's flow, structure, language, and clarity. We utilized the services of a professional editor (certificate included).
Round 4
Reviewer 3 Report
Comments and Suggestions for Authors
thanks for revision